# Digraph Inception Convolutional Networks

**Zekun Tong**[1]   **Yuxuan Liang**[2]   **Changsheng Sun**[2]   **Xinke Li**[1]
**David S. Rosenblum**[2,3]   **Andrew Lim**[1,*]

[1]Department of Industrial Systems Engineering and Management, National University of Singapore, Singapore
[2]Department of Computer Science, National University of Singapore, Singapore
[3]Department of Computer Science, George Mason University, VA, USA
`{zekuntong,liangyuxuan,cssun,xinke.li}@u.nus.edu`
`dsr@gmu.edu, isealim@nus.edu.sg`

## Abstract

Graph Convolutional Networks (GCNs) have shown promising results in modeling graph-structured data. However, they have difficulty with processing digraphs because of two reasons: 1) transforming directed to undirected graph to guarantee the symmetry of graph Laplacian is not reasonable since it not only misleads message passing scheme to aggregate incorrect weights but also deprives the unique characteristics of digraph structure; 2) due to the fixed receptive field in each layer, GCNs fail to obtain multi-scale features that can boost their performance. In this paper, we theoretically extend spectral-based graph convolution to digraphs and derive a simplified form using *personalized* PageRank. Specifically, we present the Digraph Inception Convolutional Networks (DiGCN) which utilizes digraph convolution and $k^{th}$-order proximity to achieve larger receptive fields and learn multi-scale features in digraphs. We empirically show that DiGCN can encode more structural information from digraphs than GCNs and help achieve better performance when generalized to other models. Moreover, experiments on various benchmarks demonstrate its superiority against the state-of-the-art methods.

## 1  Introduction

Learning from digraph (directed graph) data to solve practical problems, such as traffic prediction [25, 36], knowledge discovery [12] and time-series problems [4, 9], has attracted increasing attention. There are two general categories of GCNs: spatial-based [16, 40] and spectral-based [19, 23, 11]. The spatial-based approaches achieve digraph convolution by using self-defined neighbour traversal methods to aggregate features, which usually adds significant computational overhead [44, 45, 48]. Correspondingly, spectral-based GCNs [19, 44] use adjacency matrices based on spectrum analysis theory to explore neighborhood instead of traversal search, which greatly reduces the training time. However, they are limited to use undirected graphs as input by definition, the graph Laplacian needs to be symmetric [45]. How to extend spectral-based GCNs to the digraphs needs to be explored.

A majority of spectral-based GCNs transform digraphs to undirected by relaxing its direction structure [19, 44], i.e., trivially adding edges to symmetrize the adjacency matrices. It will not only mislead message passing scheme to aggregate the features with incorrect weights but also discard distinctive direction structure [43], such as irreversible time-series relationships. Besides, there are several works that learn specific structure by defining motifs [28], inheritance relationship [18] and second-order proximity[39]. However, these methods have to stipulate learning templates or rules in advance, and is not capable to deal with complex structures beyond their definitions.

---

Besides, most of the existing spectral-based GCNs enhance their capabilities of feature extraction by stacking a number of graph convolutional layers [22, 13]. However, it often leads to feature dilution as well as overfitting problem when models become deep [18, 22]. Inspired by Inception Network for image classification [37], some works [32, 46, 1] widen their layers to obtain larger receptive fields and increase learning abilities. However, they use the fixed adjacency matrix in one layer, which increases the difficulty to capture multi-scale features. A scalable neighborhood would be desirable to provide more scale information, especially for nodes belonging to communities with different sizes. Moreover, choosing a proper receptive field scheme to fuse multi-scale features together can help handle complex structures in digraphs.

To address these issues, we first extend the spectral-based graph convolution to digraphs by leveraging the inherent connections between graph Laplacian and stationary distributions of PageRank [29]. Since original digraph is not necessarily irreducible and aperiodic, the corresponding Markov chain does not have unique stationary distribution. To solve this problem, we add a chance of teleporting back to every node based on PageRank. However, the derived digraph Laplacian is too dense, and it is extremely time-consuming to perform convolution. Thus, referring to *personalized* PageRank [3], we introduce an extra auxiliary node as the teleport connected with every node to simplify fully-connected links in PageRank. The simplified digraph Laplacian can dramatically reduce the number of edges without changing the properties (irreducible and aperiodic). In addition, we theoretically analyze its properties and find that our Laplacian is the intermediate form between the undirected and directed graph, and the degree of conversion is determined by the teleport probability $\alpha$.

Moreover, inspired by the Inception Network [37], we exploit $k^{th}$-order proximity between two nodes in a digraph, which is determined through the shared $k^{th}$-order neighborhood structures of these two nodes. This does not require direct $k^{th}$-hop paths between them. By using this method, we design scalable receptive fields, which not only allows us to learn features of different sizes within one convolutional layer but also get larger receptive fields. This notion of proximity also appears in network analysis (HITS[20, 51]), psychology [30] and daily life: people who have a lot of common friends are more likely to be friends. In this way, we avoid yielding unbalanced receptive fields caused by the asymmetric paths in digraphs. Besides that, to obtain $r$-range receptive field, our model only requires stacking $\lceil \log_k r \rceil$ layers instead of $r$ GCN layers in conventional approaches.

Through experiments, we empirically show that **D**igraph **I**ncpetion **C**onvolutional **N**etworks (**DiGCN**) outperforms against competitive baselines. Additionally, our digraph convolution is superior to GCN's convolution in the mainstream directed graph benchmarks, especially over 20% accuracy on CORA-ML dataset. Our implement can be obtained at `https://github.com/flyingtango/DiGCN`.

## 2 Digraph Convolution

In this section, we first give the definition of digraph Laplacian based on PageRank, which is too dense to perform convolution well. We then simplify it by *personalized* PageRank and analyze its properties. Finally, we give the definition of digraph convolution based on the above operations.

### 2.1 Digraph Laplacian based on PageRank

Formally, given a digraph (directed graph) $\mathcal{G} = (\mathcal{V}, \mathcal{E})$, its adjacency matrix can be denoted as $\mathbf{A} = \{0,1\}^{n \times n}$, where $n = |\mathcal{V}|$. The nodes are described by the feature matrix $\mathbf{X} \in \mathbb{R}^{n \times c}$, with the number of features $c$ per node. GCN [19] proposes the *spectral graph convolution* as $\mathbf{Z}_u = \hat{\mathbf{A}}_u \mathbf{X} \mathbf{\Theta}$, where $\mathbf{Z}_u \in \mathbb{R}^{n \times d}$ is the convolved result with output dimension $d$, $\mathbf{\Theta} \in \mathbb{R}^{c \times d}$ is trainable weight and $\hat{\mathbf{A}}_u$ is the normalized self-looped version of undirected adjacency matrix $\mathbf{A}_u$ (see [19]). GCN and its variants need the **undirected** symmetric adjacency matrix $\mathbf{A}_u$ as input, therefore, they transform asymmetric $\mathbf{A}$ to symmetric form by relaxing direction structure of digraphs, e.g., let $\mathbf{A}_u = (\mathbf{A} + \mathbf{A}^T)/2$ in their experiments[1].

Noticing the inherent connections between graph Laplacian and stationary distributions of PageRank [29], we can use the properties of Markov chain to help us solve the problem in digraphs. Given a digraph $\mathcal{G} = (\mathcal{V}, \mathcal{E})$, a random walk on $\mathcal{G}$ is a Markov process with transition matrix $\mathbf{P}_{rw} = \mathbf{D}^{-1}\mathbf{A}$,

where the diagonal degree matrix $\mathbf{D}(i,i) = \sum_j \mathbf{A}(i,j)$. The $\mathcal{G}$ may contain isolated nodes in the periphery or could be formed into bipartite graph. Thus, $\mathbf{P}_{rw}$ is not necessarily *irreducible* and *aperiodic*, we can not guarantee this random walk has unique stationary distribution.

In order to relax this constraint, we slightly modify the random walk to PageRank which adds a small chance of teleporting back to every node. The PageRank transition matrix is defined as $\mathbf{P}_{pr} = (1-\alpha)\mathbf{P}_{rw} + \frac{\alpha}{n}\mathbf{1}^{n \times n}$, where $\alpha \in (0,1)$ is the teleport probability [8] and be controlled to keep the probability $\frac{\alpha}{n}$ in a small range. It is easy to prove $\mathbf{P}_{pr}$ is *irreducible* and *aperiodic*, thus, it has a unique left eigenvector $\pi_{pr}$ (also called Perron vector) which is strictly positive with eigenvalue 1 according to *Perron-Frobenius* Theory [5].

The row-vector $\pi_{pr}$ corresponds to the stationary distribution of $\mathbf{P}_{pr}$ and we have $\pi_{pr}(i) = \sum_{i, i \rightarrow j} \pi_{pr}(i)\mathbf{P}_{pr}(i,j)$. That is, the probability of finding the walk at vertex $i$ is the sum of all the incoming probabilities from vertices $j$ that have a directed edges to $i$. Thus, $\pi_{pr}$ has analogy property with nodes degree matrix $\tilde{\mathbf{D}}_u$ in undirected graph that reflecting the connectivity between nodes [15]. Using this property, we define the digraph Laplacian using PageRank $\mathcal{L}_{pr}$ in symmetric normalized format [10] as follows:

$$\mathcal{L}_{pr} = \mathbf{I} - \frac{1}{2}\left( \mathbf{\Pi}_{pr}^{\frac{1}{2}}\mathbf{P}_{pr}\mathbf{\Pi}_{pr}^{-\frac{1}{2}} + \mathbf{\Pi}_{pr}^{-\frac{1}{2}}\mathbf{P}_{pr}^T\mathbf{\Pi}_{pr}^{\frac{1}{2}} \right), \tag{1}$$

where we use $\mathbf{\Pi}_{pr} = \frac{1}{\|\pi_{pr}\|_1}\mathrm{Diag}(\pi_{pr})$ to replace $\tilde{\mathbf{D}}_u$ in undirected graph Laplacian [19]. In contrast to $\mathbf{P}_{pr}$, this matrix is symmetric. Likewise, another work [26] also employs this idea to solve digraph problem. However, it is defined on the strongly connected digraphs, which is not universally applicable to any digraphs. Our method can easily generalize to it by $\alpha \rightarrow 0$.

Adding a chance of teleporting back to every node guarantees $\pi_{pr}$ exists and makes $\mathcal{L}_{pr}$ a $\mathbb{R}^{n \times n}$ dense matrix at the same time. Using this Laplacian matrix leads to greatly increase computational overhead of convolution operation and memory requirement of $\mathcal{O}(n^2)$ for training (see time usage in Section 6). To deal with it, we propose a simplified sparse Laplacian using *personlized* PageRank.

## 2.2 Approximate Digraph Laplacian based on Personalized PageRank

To solve this issue, reconsider the equation $\mathbf{P}_{pr} = (1-\alpha)\mathbf{P}_{rw} + \frac{\alpha}{n}\mathbf{1}^{n \times n}$ of PageRank. Instead of viewing it as a combination of the random walk $\mathbf{P}_{rw}$ with a fully-connected teleport transition matrix, we can also view it as a variant of *personalized* PageRank matrix using all the nodes as teleports. To retain properties while sparse the Laplacian, we design an auxiliary node scheme.

**Using Auxiliary Node as Teleport.** More precisely, we introduce an auxiliary node $\xi \notin \mathcal{V}$ as the *personalized* PageRank teleport set $\mathcal{T} = \{\xi\}$. Based on it, we define the transition matrix of *personalized* PageRank $\mathbf{P}_{ppr}$ in the graph $\mathcal{G}_{ppr}$ as follows:

$$\mathbf{P}_{ppr} = \begin{bmatrix} (1-\alpha)\tilde{\mathbf{P}} & \alpha\mathbf{1}^{n \times 1} \\ \frac{1}{n}\mathbf{1}^{1 \times n} & 0 \end{bmatrix}, \quad \mathbf{P}_{ppr} \in \mathbb{R}^{(n+1) \times (n+1)}, \tag{2}$$

where $\tilde{\mathbf{P}} = \tilde{\mathbf{D}}^{-1}\tilde{\mathbf{A}}$, $\tilde{\mathbf{A}} = \mathbf{A} + \mathbf{I}^{n \times n}$ denotes the adjacency matrix with added self-loops and $\tilde{\mathbf{D}}(i,i) = \sum_j \tilde{\mathbf{A}}(i,j)$. Adding self-loops makes $\mathbf{P}_{ppr}$ aperiodic due to the greatest common divisor of the lengths of its cycles is one. Meanwhile, each node in $\mathcal{V}$ has a $\alpha$ possibility of linking to $\xi$ and $\xi$ has a $1/n$ possibility of teleporting back to every node in $\mathcal{V}$, which guarantees $\mathbf{P}_{ppr}$ to be irreducible. Thus, $\mathbf{P}_{ppr}$ has a unique left eigenvector $\pi_{ppr} \in \mathbb{R}^{n+1}$ which is strictly positive with eigenvalue 1.

**Approximate Digraph Laplacian.** Our target is finding the Laplacian of $\tilde{\mathbf{P}}$ for spectral analysis, however, $\tilde{\mathbf{P}}$ is not necessarily irreducible, which means the eigenvector $\tilde{\pi} \in \mathbb{R}^n$ with the largest eigenvalue of $\tilde{\mathbf{P}}$ is not unique. Thus, we use the stationary distribution of $\mathbf{P}_{ppr}$ to approximate the stationary distribution of $\tilde{\mathbf{P}}$. We can split $\pi_{ppr}$ into two parts: $\pi_{ppr} = (\pi_{appr}, \pi_\xi)$, where $\pi_{appr} \in \mathbb{R}^n$ is the unique stationary distribution of the first $n$ points and $\pi_\xi \in \mathbb{R}^1$ is the unique stationary distribution of the auxiliary node $\xi$. $\pi_{appr}$ can converge to stationary distribution of $\tilde{\mathbf{P}}$ according to THEOREM 1.

**THEOREM 1** *Based on the definitions, when teleport probability $\alpha \rightarrow 0$, $\pi_{appr}\tilde{\mathbf{P}} - \pi_{appr} \rightarrow 0$.*

We can control $\alpha$ in a small range then simplify Equation 1 to:

$$\mathcal{L}_{appr} = \mathbf{I} - \frac{1}{2}\left(\tilde{\mathbf{\Pi}}^{\frac{1}{2}}\tilde{\mathbf{P}}\tilde{\mathbf{\Pi}}^{-\frac{1}{2}} + \tilde{\mathbf{\Pi}}^{-\frac{1}{2}}\tilde{\mathbf{P}}^T\tilde{\mathbf{\Pi}}^{\frac{1}{2}}\right) \approx \mathbf{I} - \frac{1}{2}\left(\mathbf{\Pi}_{appr}^{\frac{1}{2}}\tilde{\mathbf{P}}\mathbf{\Pi}_{appr}^{-\frac{1}{2}} + \mathbf{\Pi}_{appr}^{-\frac{1}{2}}\tilde{\mathbf{P}}^T\mathbf{\Pi}_{appr}^{\frac{1}{2}}\right), \quad (3)$$

where $\tilde{\mathbf{\Pi}} = \frac{1}{\|\tilde{\pi}\|_1}\operatorname{Diag}(\tilde{\pi})$ and $\mathbf{\Pi}_{appr} = \frac{1}{\|\pi_{appr}\|_1}\operatorname{Diag}(\pi_{appr})$. Note that $\mathcal{L}_{appr}$ retains the graph's sparsity of $\mathcal{O}(|\mathcal{E}|)$. Next, we give the conditions to converge to other forms in THEOREM 2.

**THEOREM 2** *Based on the above definitions, given an input graph $\mathcal{G}$, when teleport probability $\alpha \to 1$, $\mathbf{\Pi}_{appr} \to \frac{1}{n}\cdot\mathbf{I}^{n\times n}$ and the approximate digraph Laplacian converges as $\mathcal{L}_{appr} \to \mathbf{I} - \frac{1}{2}(\tilde{\mathbf{P}} + \tilde{\mathbf{P}}^T)$, which is a trivial-symmetric Laplacian matrix. Specially, if $\mathcal{G}$ is undirected, then the approximate digraph Laplacian converges as $\mathcal{L}_{appr} \to \mathbf{I} - \tilde{\mathbf{D}}^{-1}\tilde{\mathbf{A}}$, which is a random-walk normalized Laplacian.*

**Generalization.** We show in THEOREM 2 that two common used undirected Laplacian matrices are special cases of our method under certain conditions: the trivial-symmetric one mentioned in Section 2.1 and random-walk normalized one. When $\alpha$ tends to 1, the form of our method is closer to the form of undirected graph Laplacian. That is to say $\alpha$ can control the degree of conversion from a directed form to an undirected form. The smaller $\alpha$ retains the more directed properties, and vice versa. Proofs of THEOREM 1 and 2 are attached in the Supplementary Material.

## 2.3 Digraph Convolution

As we have defined the digraph Laplacian in Equation 3 and it is symmetric, we can follow the spectral analysis in GCN [19] to derive the definition of the digraph convolution as:

$$\mathbf{Z} = \frac{1}{2}\left(\mathbf{\Pi}_{appr}^{\frac{1}{2}}\tilde{\mathbf{P}}\mathbf{\Pi}_{appr}^{-\frac{1}{2}} + \mathbf{\Pi}_{appr}^{-\frac{1}{2}}\tilde{\mathbf{P}}^T\mathbf{\Pi}_{appr}^{\frac{1}{2}}\right)\mathbf{X}\mathbf{\Theta}, \quad (4)$$

where $\mathbf{Z} \in \mathbb{R}^{n\times d}$ is the convolved result with $d$ output dimension, $\mathbf{X} \in \mathbb{R}^{n\times c}$ is node feature matrix and $\mathbf{\Theta} \in \mathbb{R}^{c\times d}$ is trainable weight. Note that we carry out row normalization to the input weighted adjacency matrix. This propagation scheme has complexity $\mathcal{O}(|\mathcal{E}|cd)$ which is same with GCN [19], as digraph Laplacian is sparse and can be calculated during preprocessing.

# 3 Digraph Inception Network

In this section, first, we introduce $k^{th}$-order Proximity as scalable receptive field and then we present **DiGCN**, a multi-scale inception network, to learn from features of different size in digraphs.

## 3.1 Scalable Receptive Field for Digraph based on $k^{th}$-order Proximity

We start by explaining the feature spreading ways in GCNs. Xu et al. [47] have shown that the information of node $i$ spreads to node $j$ in an analogous random walk manner, which means path is the way of feature transmission and the size of receptive field is determined by the length of the path in a graph. However, in digraph, long paths only exist between a few points and are often not bidirectional, which is not conducive to obtaining global features. Meanwhile, different communities have various node degrees of in and out, which may cause unbalanced receptive fields (paths) in digraphs. To solve this problem, we propose $k^{th}$-order Proximity in digraphs which not only obtains the node's features from its directly adjacent nodes, but also extract the hidden information from $k^{th}$-order neighbor nodes. That is, if two nodes share common neighhors, they tend to be similar.

**DEFINITION 1** $k^{th}$-***order Proximity***. *Given a graph $\mathcal{G} = (\mathcal{V}, \mathcal{E})$, for $k \geqslant 2$, if $\exists\, e \in \mathcal{V}$ and a path between node $i$ and $j$ ( $i, j \in \mathcal{V}$) in this form: $v_i \underbrace{\to\cdots\to}_{k-1\ edges} v_e \underbrace{\leftarrow\cdots\leftarrow}_{k-1\ edges} v_j$, we define this path as $k^{th}$-order meeting path $\mathcal{M}_{i,j}^{(k)}$. Similarity, the $k^{th}$-order diffusion path $\mathcal{D}_{i,j}^{(k)}$ is $v_i \underbrace{\leftarrow\cdots\leftarrow}_{k-1\ edges} v_e \underbrace{\to\cdots\to}_{k-1\ edges} v_j$. If there exist both $\mathcal{M}_{i,j}^{(k)}$ and $\mathcal{D}_{i,j}^{(k)}$ between node $i$ and $j$, we think they are $k^{th}$-order proximity and $e$ is their $k^{th}$-order common neighbor. Note that one node is $0^{th}$-order proximity with itself and $1^{st}$-order proximity with its directly connected neighbors.*

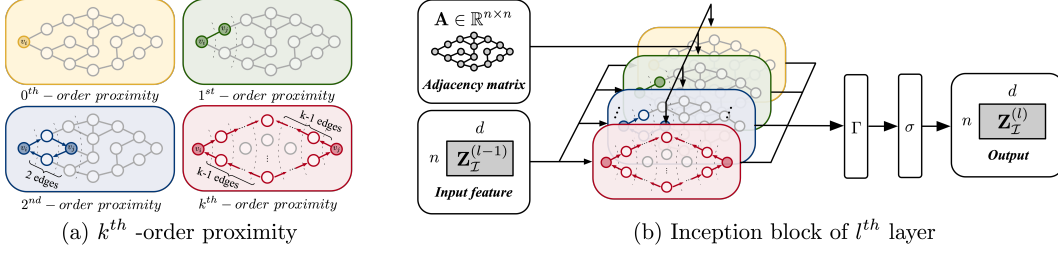

(a) $k^{th}$ -order proximity

(b) Inception block of $l^{th}$ layer

Figure 1: Illustration of DiGCN model. For a digraph $\mathcal{G}$, we use $k^{th}$-order proximity to generate $k$ receptive fields based on the input adjacency matrix $\mathbf{A}$ shown in the Subfigure (a). Then we convolute them with input feature matrix $\mathbf{Z}_{\mathcal{I}}^{(l-1)}$ and gain the output $\mathbf{Z}_{\mathcal{I}}^{(l)}$ after fusion. We encapsulate this process as an Inception block shown in the Subfigure (b), where $l \in \mathbb{Z}^+$ is the number of layers and the initial input $\mathbf{Z}_{\mathcal{I}}^{(0)} = \mathbf{X}$. Multi-layer networks can be implemented by stacking Inception blocks.

The schematic of $k^{th}$ -order proximity shows in Figure 1(a). Based on Definition 1, we build a $k^{th}$-order proximity matrix to connect similar nodes together.

**DEFINITION 2** $k^{th}$**-order Proximity Matrix.** *In order to model the $k^{th}$-order proximity, we define the $k^{th}$-order proximity matrix $\mathbf{P}^{(k)}(k \in \mathbb{Z})$ in the graph $\mathcal{G}$:*

$$\mathbf{P}^{(k)} = \begin{cases} \mathbf{I} & k = 0 \\ \tilde{\mathbf{D}}^{-1}\tilde{\mathbf{A}} & k = 1 \\ \text{Intersect}\left((\mathbf{P}^{(1)})^{k-1}(\mathbf{P}^{(1)^T})^{k-1}, (\mathbf{P}^{(1)^T})^{k-1}(\mathbf{P}^{(1)})^{k-1}\right)/2 & k \geqslant 2 \end{cases}. \quad (5)$$

$\tilde{\mathbf{A}}$ *is the adjacency matrix with self-loops of $\mathcal{G}$ and $\tilde{\mathbf{D}}$ is corresponding diagonalized degree matrix.* $\text{Intersect}(\cdot)$ *denotes the element-wise intersection of matrices that only when the corresponding positions have both meeting and diffusion paths, the sum operation is performed, otherwise, it is 0.*

The $k^{th}$-order proximity matrix $\mathbf{P}^{(k)}$ is symmetric if $k \geqslant 2$ because of the intersection operation. $k$ represents distance between two similar nodes, that is, the size of the receptive fields. We can get scalable receptive fields by setting different $k$.

## 3.2 Multi-scale Inception Network Structure

Based on the proposed $k^{th}$-order proximity matrix, we define the multi-scale digraph convolution as:

$$\mathbf{Z}^{(k)} = \begin{cases} \mathbf{X}\Theta^{(0)} & k = 0 \\ \frac{1}{2}\left(\mathbf{\Pi}^{(1)\frac{1}{2}}\mathbf{P}^{(1)}\mathbf{\Pi}^{(1)-\frac{1}{2}} + \mathbf{\Pi}^{(1)-\frac{1}{2}}\mathbf{P}^{(1)^T}\mathbf{\Pi}^{(1)\frac{1}{2}}\right)\mathbf{X}\Theta^{(1)} & k = 1 \\ \mathbf{W}^{(k)-\frac{1}{2}}\mathbf{P}^{(k)}\mathbf{W}^{(k)-\frac{1}{2}}\mathbf{X}\Theta^{(k)} & k \geqslant 2 \end{cases}. \quad (6)$$

$\mathbf{Z}^{(k)} \in \mathbb{R}^{n \times d}$ are convolved results with $d$ output dimension, $\mathbf{X} \in \mathbb{R}^{n \times c}$ is node feature matrix, $\mathbf{W}^{(k)}$ is diagonalized weight matrix of $\mathbf{P}^{(k)}$ and $\Theta^{(k)} \in \mathbb{R}^{c \times d}$ is trainable weight. Note that when $k = 1$, $\mathbf{Z}^{(1)}$ is calculated by digraph convolution with $\mathbf{P}^{(1)}$ in Section 2.3 and $\mathbf{\Pi}^{(1)}$ is the corresponding approximate diagonalized eigenvector.

Inspired by the **Inception module** proposed in [37], we build the multi-scale digraph Inception network. We can compare $\mathbf{P}^{(k=0)}$ to $1 \times 1$ convolution kernel and treat $\mathbf{Z}^{(k=0)}$ as a skip connection term carrying non-smoothed features. Moreover, $\mathbf{Z}^{(k \geqslant 1)}$ is designed to encode multi-scale directed structure features. Finally, we use fusion operation $\Gamma$ to fusion multi-scale features together as an Inception block $\mathbf{Z}_{\mathcal{I}}$:

$$\mathbf{Z}_{\mathcal{I}} = \sigma(\Gamma(\mathbf{Z}^{(0)}, \mathbf{Z}^{(1)}, ..., \mathbf{Z}^{(k)})), \quad (7)$$

where $\sigma$ is activation function. Fusion function $\Gamma$ can be various, such as normalization, summation and concatenation. In practice, we use $\Gamma$ to keep the feature dimensions unchanged, that is keeping $\mathbf{Z}_{\mathcal{I}} \in \mathbb{R}^{n \times d}$ for stacking the same block. The schematic of Inception block shows in Figure 1(b). We notice that a recent work SIGN uses a similar Inception structure to handle large scale graph learning [32]. Differently, they use SGC [44] as basic block which is not applicable to digraphs and

concatenate these block of different size together into a FC layer. However, concatenation is a kind of features fusion method, and in some cases, the effect of concatenation is not as good as summation, we illustrate this in Section 6.

**Generalization to other Models.** Our method using $k^{th}$-order proximity to improve the convolution receptive field has strong generalization ability. In most spectral-based models, we can use our Inception block to replace the original layer (see Table 1).

Table 1: Our Inception block can generalizate to other models only need to modify some parameters. Here, $\mathbf{\Lambda}$ is Laplacian eigenvalue matrix defined in ChebNet [11] and $\mathbf{A}_u$ is the symmetric form of $\mathbf{A}$.

|  | Undigraph | Digraph | Adj | Scale Range | Weights | Fusion Method |
|---|---|---|---|---|---|---|
| ChebNet [11] | ✓ |  | $\mathbf{\Lambda}$ | $[0, 1, .., k]$ | $\mathbf{\Theta}^0, ..., \mathbf{\Theta}^k$ | Sum |
| GCN [19] | ✓ |  | $\mathbf{A}_u$ | 1 | $\mathbf{\Theta}$ | None |
| SGC [44] | ✓ |  | $\mathbf{A}_u$ | $k$ | $\mathbf{\Theta}$ | None |
| SIGN [32] | ✓ |  | $\mathbf{A}_u$ | $[0, 1, .., k]$ | $\mathbf{\Theta}^0, ..., \mathbf{\Theta}^k$ | Concate |
| **Ours (DiGCN)** | ✓ | ✓ | $\mathbf{A}$ | $[0, 1, .., k]$ | $\mathbf{\Theta}^0, ..., \mathbf{\Theta}^k$ | Any $\Gamma$ |

Taking SGC [44] as an example, we can generalize our method to the SGC by replacing the origin $k^{th}$-power of adjacency matrix by Inception block. Experimental results in Section 6 show that integrating our method can help improve accuracy.

**Time and Space Complexity.** For digraph convolution defined in the Equation 4, we can use a sparse matrix to store Laplacians. And as we use full batch training, the memory space cost for one adjacency matrix is $\mathcal{O}(|\mathcal{E}|)$.

For Inception block defined in the Equation 7, due to the asymmetry of the digraphs mentioned in Section 3.1, long paths normally exist between a few points and are not bidirectional. Thus, using $k^{th}$-order proximity will get unbalanced receptive field and introduce computational complexity. Intersection and union of meeting and diffusion paths both can handle unbalancing problem. We compare the number of edges per Inception block with different $k$ on CORA-ML [6] and CITESEER [33] shown in Figure 2 and find that the edges in $k^{th}$-order matrix will not increase exponentially and intersection does help to reduce memory consume. Thus, the memory space cost for one Inception block in practical is $\mathcal{O}(k|\mathcal{E}|)$. However, the worst

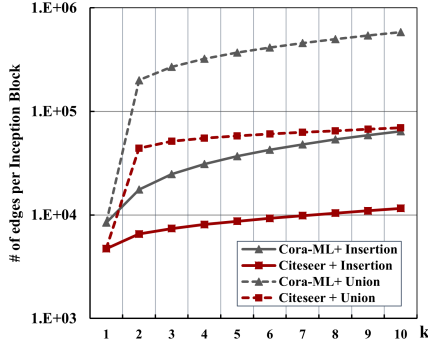

Figure 2: # of edges per Inception block

case does exist when the input graph is undirected and strongly connected. Though it is unsuitable to our model, which mainly treats reducible digraphs, the worst space complexity is $\mathcal{O}(k|\mathcal{V}|^2)$.

We can calculate eigenvalue decomposition in the Equation 3 during the preprocessing and store the results, therefore the computational complexity is $\mathcal{O}(|\mathcal{V}|^3)$. At the same time, we use the sparse matrix multiplication. Thus, we can obtain the complexity of convolution as $\mathcal{O}(k|\mathcal{E}|cd)$.

## 4 Related Work

**Digraph Convolution.** Several works have tried to make GCNs adaptive to digraphs by looking for structural patterns and reformulate the graph [28, 39, 18]. However, these methods have their limitations that rely on pre-defined structure examples and can not handle complex structure which do not appear in the patterns. An alternative method [26] redefines the propagation scheme from Markov process view, which only apply for strongly connected digraph according to their definition. Our approach is universally applicable to any digraphs, which is the biggest difference from them.

**PageRank in GCNs.** Klicpera et al. [21] propose PPNP model which utilize a propagation based on personalized PageRank to handle feature oversmoothing problem. Besides, the PPRGo [7] increases the efficiency of PPNP by incorporating multi-hop neighborhood information in a single step. Note that no matter PPNP or PPRGo, the basic form of their propagation matrices is $\mathbf{A}_{ppnp} = \alpha \left( \mathbf{I} - (1 - \alpha) \mathbf{A}_u \right)^{-1}$, which is quite different from our digraph Laplacian in Equation 3. Furthermore, they use symmetric matrix $\mathbf{A}_u$ in the propagation, which means they are not adaptive to digraphs.

$k^{th}$-**order Proximity and Multi-scale Receptive Field.** Previous works have found the powerful information extraction capabilities of $k^{th}$-order proximity [38, 38, 51, 39]. However, they only consider first- and second-order proximity to find hidden information, and our method can adjust the proximity range as needed. There are several methods to achieve multi-scale receptive field, such as k-hop method [2, 11, 1, 44] and graph inception [32, 46, 31]. In our model, we use $k^{th}$-order proximity instead of the k-hop method because they may create unbalanced receptive filed in digraphs (see Section 3.1). And the existing graph inception models are not suitable to digraphs (see Table 1).

## 5   Experimental Settings

We conduct extensive experiments to evaluate the effectiveness of our model. The Supplementary Material reports further details on the experiments and reproducibility.

**Experimental Task.** Node classification is a common task used to measure graph models. In this paper, we adopt the task of **Semi-supervised Node Classification in Digraphs** to verify the learning ability of models. Compared with the common experiments for undirected graphs [19, 44, 21, 42], the challenge is that the given adjacency matrix $\mathbf{A}$ is asymmetric, which means message passing has its direction. Based on the method proposed above, we build several simple models to deal with this problem. Task definition and specific model structures, including schematics, loss function and configuration, are included in the Supplementary Material.

**Baselines.** We compare our model to eight state-of-the-art models that can be divided into four main categories: 1) **spectral-based** GNNs including ChebNet [11], GCN [19], SGC [44] , APPNP [21] and InfoMax [41]; 2) **spatial-based** GNNs containing GraphSage [16] and GAT [40]; 3) **Digraph** GNNs including DGCN [39] (we do not use [26] because it only apply for strongly connected graph which needs cropping the original dataset to meet its settings); 4) **Graph Inception** having SIGN [32]. The descriptions and settings of them are introduced in the Supplementary Material.

**Datasets and Splitting.** We use several **digraph** datasets including citation networks: CORA-ML [6] and CITESEER [33], and Amazon Co-purchase Networks: AM-PHOTO and AM-COMPUTER [34]. The split of the datasets will greatly affect the performance of the models [34, 21]. Especially for a single split, not only will it cause overfitting problems during training, but it is also easy to get misleading results. Thus, in our experiments, we randomly split the datasets and perform multiple experiments to obtain stable and reliable results. For train/validation/test split, following the rules in GCN [19], we choose 20 labels per class for training set, 500 labels for validation set and rest for test set. The detailed descriptions are summarized in the Supplementary Material.

## 6   Experimental Results

**Overall accuracy.** The performance comparisons between our model and baselines on four datasets are reported in Table 2. We train all models for a maximum of 1000 epochs and early stop if the validation accuracy does not increase for 200 consecutive epochs, then calculate mean test accuracy with STD in percent (%) averaged over 20 random dataset splits with random weight initialization.

It can be seen easily that our methods achieves the state-of-the-art results on all datasets. Notice that spectral-based models including ChebNet, GCN, SGC and InfoMax, do not perform well on digraph datasets compared to their good performance in undirected graphs. This is mainly because these models have limited ability to obtain features from the surroundings using asymmetric adjacency matrices. However, APPNP is an exception. It allows features to randomly propagate with a certain teleport probability, which breaks through the path limitation and achieves good results in digraphs. The spatial-based method and ours have similar results, which shows that both methods have good suitability for digraphs. Moreover, DGCN performs well on the most datasets, however, it uses both in- & out-degree proximity matrix to obtain structural features in digraphs, which leads to out of memory on AM-COMPUTER. Meanwhile, SIGN uses SGC as the basic module, thus, even if Inception method is used, SIGN does not perform well in digraphs (see analysis in Section 3.2).

**Training time.** With the same training settings, we measure the convergence speed of models by average training time per run in second (s) in Table 2. Apparently, our models have similar results. $\mathcal{L}_{pr}$ can only be applied to moderately sized graphs, while $\mathcal{L}_{appr}$ scales to large graphs. Compared with the spectral-based methods, the overall speed of our model without Inception block is similar to

Table 2: Overall accuracy and training time. "w/ pr" means using $\mathcal{L}_{pr}$; "w/ appr" means using $\mathcal{L}_{appr}$; "w/o IB" means using digraph convolution only; "w/ IB" means using Inception block. The best results are highlighted with **bold** and the second are highlighted with underline.

| Model | CORA-ML | | CITESEER | | AM-PHOTO | | AM-COMPUTER | |
|---|---|---|---|---|---|---|---|---|
| | acc | time | acc | time | acc | time | acc | time |
| ChebNet [11] | $64.02 \pm 1.5$ | 7.23 | $56.46 \pm 1.4$ | 7.45 | $80.91 \pm 1.0$ | 10.52 | $73.25 \pm 0.8$ | 16.96 |
| GCN [19] | $53.11 \pm 0.8$ | 4.48 | $54.36 \pm 0.5$ | 4.80 | $53.20 \pm 0.4$ | 4.86 | $60.50 \pm 1.6$ | 5.04 |
| SGC [44] | $51.14 \pm 0.6$ | 1.92 | $44.07 \pm 3.5$ | 3.58 | $71.25 \pm 1.3$ | 2.31 | $76.17 \pm 0.1$ | 3.68 |
| APPNP [21] | $70.07 \pm 1.1$ | 6.84 | $65.39 \pm 0.9$ | 6.94 | $79.37 \pm 0.9$ | 6.72 | $63.16 \pm 1.4$ | 6.47 |
| InfoMax [41] | $58.00 \pm 2.4$ | 4.11 | $60.51 \pm 1.7$ | 4.85 | $74.40 \pm 1.2$ | 31.80 | $47.32 \pm 0.7$ | 41.96 |
| GraphSage [16] | $72.06 \pm 0.9$ | 6.22 | $63.19 \pm 0.7$ | 6.21 | $87.57 \pm 0.9$ | 8.52 | $79.29 \pm 1.3$ | 14.49 |
| GAT [40] | $71.91 \pm 0.9$ | 6.02 | $63.03 \pm 0.6$ | 6.12 | $89.10 \pm 0.7$ | 8.83 | $79.45 \pm 1.5$ | 14.66 |
| DGCN [39] | $75.02 \pm 0.5$ | 6.53 | $66.00 \pm 0.4$ | 6.84 | $83.66 \pm 0.8$ | 36.29 | OOM | - |
| SIGN [32] | $66.47 \pm 0.9$ | 2.81 | $60.69 \pm 0.4$ | 2.96 | $74.13 \pm 1.0$ | 5.33 | $69.40 \pm 4.8$ | 4.97 |
| Ours | | | | | | | | |
| w/ pr w/o IB | $77.11 \pm 0.5$ | 39.13 | $64.77 \pm 0.6$ | 47.19 | OOM* | - | OOM | - |
| w/ appr w/o IB | $77.01 \pm 0.4$ | 2.71 | $64.92 \pm 0.3$ | 2.69 | $88.72 \pm 0.3$ | 2.95 | $85.55 \pm 0.4$ | 4.23 |
| w/ appr w/ IB | **$80.28 \pm 0.5$** | 6.38 | **$66.11 \pm 0.7$** | 6.42 | **$90.02 \pm 0.5$** | 11.77 | **$85.94 \pm 0.5$** | 26.63 |

* OOM stands for out of memory (see efficiency analysis in Section 2.1)

SGC since the Laplacian is precomputed. On AM-PHOTO and AM-COMPUTER with large scales, our model is 30% faster on average than GCN. The accuracy of our model improves significantly while the speed decreases after using Inception, which is consistent with complexity analysis in Section 3.2.

**Ablation study.** We validate the effectiveness of the components and the resulting ACC are shown in Table 2. Comparing model with $\mathcal{L}_{appr}$ and model with $\mathcal{L}_{pr}$, we find that the approximate method can not only achieve the similar accuracy but also save training time and memory. Meanwhile, we find that the combination of $\mathcal{L}_{appr}$ and Inception block brings significant improvement in accuracy. This substantially validate that scalable receptive fields do help to learn features from neighborhood.

**Effect of teleport probability $\alpha$.** Figure 3(a) shows the effect of the hyperparameter $\alpha$ on the test accuracy and structural retention. Referring to the assumption in Section 2.2, we define structural retention as $\mathcal{S} = \text{KL}(\pi_{appr}, \pi_{appr}\tilde{\mathbf{P}})^{-1}$, where KL means KL divergence. We use $\mathcal{S}$ to measure how much directed structural information retained after approximation. The smaller $\mathcal{S}$, the less information remain. According to Theorem 1 that $\alpha$ needs to be close to 0 to retain digraph structural information in Laplacian, however, we find that a higher $\alpha$ improves accuracy slightly. In view of this, we choose $\alpha \in [0.05, 0.2]$ to balance structural retention and accuracy. $\alpha$ should be adjusted for the different datasets [21], but in order to maintain consistency, we take $\alpha = 0.1$ in all experiments.

**Link prediction in digraphs.** We use link prediction task in digraphs to show that our model is able to obtain more structural information. In this task, we split the edges of a digraph into positive and negative train/val/test edges and compare the results with GCN over 20 runs for a maximum of 500 epochs. Figure 3(b) shows that that our model (w/ appr and w/o IB) outperforms GCN for link prediction on all datasets. This is mainly because we take the direction of the edges into account when calculating $\mathcal{L}_{appr}$, which allows us to obtain more accurate structural information in digraphs.

**Training time at different graph scales.** We report results for the mean training time in millisecond (ms) per epoch for 200 epochs on simulated random graphs using digraph convolution. We construct a simple random graph with $N$ nodes and assign $2N$ edges uniformly at random. We take the node index matrix as input feature matrix and give the same label for every node. Figure 3(c) summarizes the results and shows that our model can handle about 10 million nodes in one GPU (11GB).

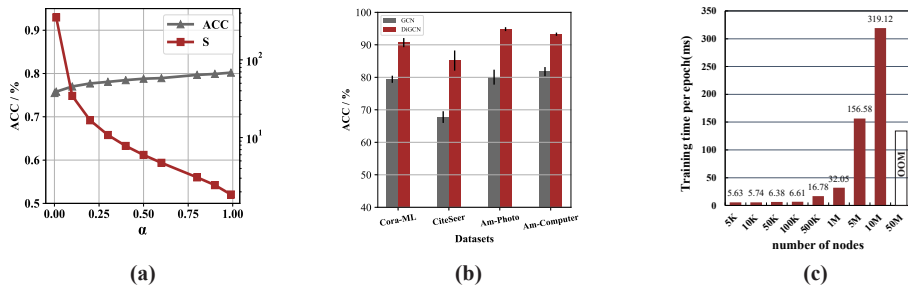

(a)　　　　　　　　　(b)　　　　　　　　　(c)

Figure 3: (a) effect of $\alpha$ to ACC and structural retention $\mathcal{S}$; (b) link prediction results on different datasets; (c) training time per epoch on random digraphs with difference size.

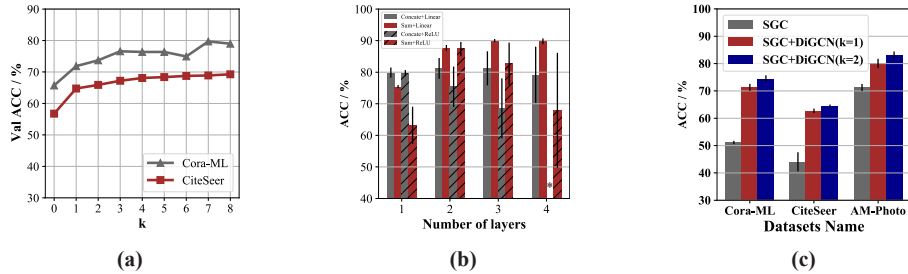

Figure 4: (a) effects of model width $k$ ; (b) fusion and activation functions on AM-PHOTO in different layers (* stands for unable to converge); (c) generalization to SGC using different $k$.

**Depth and width of Inception block.** How to balance the model depth and width becomes a vital issue for Inception block. Figure 4(a) shows that for a single layer model, the improvement in val accuracy is not significant for $k > 2$, which means larger receptive field cannot obtain more effective information on small-scale dataset CITESEER. Then, we keep $k = 2$ and carry out grid search on model depth and choose layer=3. To intuitively compare the impact of depth under the same receptive range, we choose GAT as baseline, which can obtain various range by adjusting the head size without stacking layers. From Table 3, we can observe that larger receptive fields help our model to perform better. It is consistent that using a moderate number of layers is enough to effectively learn features.

Table 3: Results under various depths. Our model sets $k = 2$, and uses $\mathcal{L}_{appr}$ as Laplacian and Sum as fusion operation. "Range" means range of receptive field. The best results are highlighted with **bold**.

| Methods | Range | Settings | CORA-ML | CITESEER | AM-PHOTO | AM-COMPUTER |
|---|---|---|---|---|---|---|
| GAT [40] | 2 | head=2 | $71.33 \pm 1.4$ | $\mathbf{63.33 \pm 0.7}$ | $\mathbf{81.12 \pm 1.5}$ | $\mathbf{75.12 \pm 3.2}$ |
| Ours | | layer=1 | $\mathbf{72.14 \pm 1.0}$ | $62.89 \pm 0.4$ | $75.43 \pm 0.5$ | $64.17 \pm 0.5$ |
| GAT [40] | 4 | head=4 | $71.65 \pm 1.0$ | $63.30 \pm 0.7$ | $86.03 \pm 1.0$ | $77.57 \pm 1.5$ |
| Ours | | layer=2 | $\mathbf{76.62 \pm 0.5}$ | $\mathbf{63.98 \pm 0.5}$ | $\mathbf{87.71 \pm 0.9}$ | $\mathbf{82.36 \pm 0.7}$ |
| GAT [40] | 8 | head=8 | $71.62 \pm 0.8$ | $63.17 \pm 0.6$ | $87.04 \pm 1.0$ | $78.22 \pm 1.7$ |
| Ours | | layer=3 | $\mathbf{80.28 \pm 0.5}$ | $\mathbf{66.11 \pm 0.7}$ | $\mathbf{90.02 \pm 0.5}$ | $\mathbf{85.94 \pm 0.5}$ |
| GAT [40] | 16 | head=16 | $71.91 \pm 0.9$ | $63.03 \pm 0.6$ | $\mathbf{89.10 \pm 0.7}$ | $79.45 \pm 1.5$ |
| Ours | | layer=4 | $\mathbf{79.95 \pm 0.8}$ | $\mathbf{64.00 \pm 1.0}$ | $89.81 \pm 0.9$ | $\mathbf{83.36 \pm 0.7}$ |

**Fusion operation and activation function.** We show results in Figure 4(b) and use Sum and Concate to represent Summation and Concatenation respectively. We find that the choices of fusion and activation functions need to match the complexity of the model. When the model is shallow, Concate performs better due to more parameters. Sum achieves better results in deep model because it requires fewer parameters, which helps prevent the model from overfitting. Since we use $k^{th}$-order proximate IB, linear combinations can achieve stable results on smaller datasets. Using a non-linear activation function (ReLU) may result in models that are too complex to learn features effectively.

**Generalization to other model (SGC).** To test the generalization ability of our Inception block, we use it to replace the origin layer in SGC [44]. The generalized SGC model is denoted by **SGC+DiGCN**. In addition, we test the case of $k = 1$ and $k = 2$ to the generalized model and the results are shown in Figure 4(c). Obviously, whether $k = 1$ or $k = 2$ our generalized model outperforms the original model on all datasets, which shows that the multi-scale receptive field helps the model obtain more surrounding information. Meanwhile, our method has good generalization ability because of its simple structure that can be plugged into existing models easily.

# 7   Conclusion and Future Work

In this paper, we present a novel Digraph Inception Convolutional Networks (**DiGCN**), which can effectively learn digraph representation. We theoretically extend spectral-based graph convolution to digraph and further simplify it. Besides, we define $k^{th}$-*order proximity* and design the digraph Inception networks to learn multi-scale features. This simple and scalable model can not only learn digraph structure, but also get hidden information through $k^{th}$-order proximity relationship. Finally, we use several tasks on various real-world datasets to validate the effectiveness and generalization capability of our model. The results show that our model outperforms several state-of-the-art methods.

Due to the full batch training, our model can not be applied to large-scale graphs and we will consider adapting it to mini-batch training in the future. In addition, we will study how to combine our model with existing models to solve more complex tasks, e.g., computer vision and NLP.

## Broader Impact

GCNs could be applied to a wide range of applications, including image segmentation [27], speech recognition [14], recommender system [17], point cloud [50, 24], traffic prediction [25] and many more [45]. Our method can help to expand the graph types from undirected to directed in the above application scenarios and obtain multi-scale features from the high-order hidden directed structure.

For traffic prediction, our method can be used in map applications to obtain more fine-grained and accurate predictions. This requires users to provide location information, which has a risk of privacy leakage. The same concerns also arise in social network analysis [38], person re-ID [35] and NLP [49], which use graph convolutional networks as their feature extraction methods. Another potential risk is that our model may be adversarial attacked by adding new nodes or deleting existing edges. For example, in a graph-based recommender system, our model may produce completely different recommendation results due to being attacked.

We see opportunities for research applying DiGCN to beneficial purposes, such as investigating the ability of DiGCN to discover hidden complex directed structure, the limitation of approximate method based on *personalized* PageRank and the feature oversmoothing problem in digraphs. We also encourage follow-up research to design derivative methods for different tasks based on our method.

## Acknowledgments and Disclosure of Funding

The authors would like to thank Jinyang Wang, Miqi Wu and Yongxing Dai for the great help in discussion and proofreading. This research is supported in part by the following grants: MOE2019-T3-1-010, MOE2017-T2-2-153, NRF-RSS2016-004.

## Footnotes

[1]There are various ways to construct an undirected graph from a digraph. In this paper, we consider one of the most commonly used methods that averaging edge weights when combination.

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
