[Supplementary Material]

# Supplementary Material for the Paper: Digraph Inception Convolutional Networks

Zekun Tong[1]   Yuxuan Liang[2]   Changsheng Sun[2]   Xinke Li[1]
David S. Rosenblum[2,3]   Andrew Lim[1,*]

[1]Department of Industrial Systems Engineering and Management, National University of Singapore, Singapore
[2]Department of Computer Science, National University of Singapore, Singapore
[3]Department of Computer Science, George Mason University, VA, USA
{zekuntong,liangyuxuan,cssun,xinke.li}@u.nus.edu
dsr@gmu.edu, isealim@nus.edu.sg

## 1   Proof that $\mathbf{P}_{pr}$ and $\mathbf{P}_{ppr}$ are Irreducible and Aperiodic

**DEFINITION 1.** *Irreducible and Aperiodic: Given an input $\mathcal{G} = (\mathcal{V}, \mathcal{E})$, $\mathcal{G}$ is irreducible iff for any two vertices $v_i, v_j \in \mathcal{V}$, there is an integer $k \in \mathbb{Z}^+$, s.t. $A_{ij}^k > 0$. Meanwhile, $\mathcal{G}$ is aperiodic iff the greatest common divisor of the lengths of its cycles is one. The random walk $\mathbf{P}$ defined on $\mathcal{G}$ has the same irreducible and aperiodic properties with $\mathcal{G}$.*

*Proof.* For $\mathbf{P}_{pr}$, since it has a $\frac{\alpha}{n}$ probability to jump from any point in $\mathcal{V}$ to another point, its corresponding graph is strongly connected and the greatest common divisor of the lengths of graph's cycles is 1. Thus, $\mathbf{P}_{pr}$ is irreducible and aperiodic. For $\mathbf{P}_{ppr}$, since it has a auxiliary node $\xi$ which is connected with every node in $\mathcal{V}$, its corresponding graph is strongly connected. Meanwhile, adding self-loops makes the greatest common divisor of the lengths of graph's cycles is 1. Thus, $\mathbf{P}_{ppr}$ is irreducible and aperiodic. $\qquad\square$

## 2   Proofs of Theorems

**THEOREM 1.** *Based on the definitions in the paper, given an input graph $\mathcal{G}$ and its personalized PageRank $\mathbf{P}_{ppr}$, when teleport probability $\alpha \to 0$, $\pi_{appr}\tilde{\mathbf{P}} - \pi_{appr} \to 0$.*

*Proof.* We start out proof from equation $\pi_{ppr}\mathbf{P}_{ppr} = \pi_{ppr}$, leading to

$$\begin{bmatrix} \pi_{appr} & \pi_\xi \end{bmatrix} \begin{bmatrix} (1-\alpha)\tilde{\mathbf{P}} & \alpha\mathbf{1}^{n\times 1} \\ \frac{1}{n}\mathbf{1}^{1\times n} & 0 \end{bmatrix} = \begin{bmatrix} \pi_{appr} & \pi_\xi \end{bmatrix}. \tag{1}$$

Therefore,

$$(1-\alpha)\pi_{appr}\tilde{\mathbf{P}} + \frac{1}{n}\pi_\xi\mathbf{1}^{1\times n} = \pi_{appr}, \tag{2}$$
$$\alpha\pi_{appr}\mathbf{1}^{n\times 1} = \pi_\xi$$

then,

$$(1-\alpha)\pi_{appr}\tilde{\mathbf{P}} + \frac{\alpha}{n}\pi_{appr} = \pi_{appr} \tag{3}$$

and

$$\pi_{appr}\tilde{\mathbf{P}} - \pi_{appr} = \frac{n-1}{n}\frac{\alpha}{1-\alpha}\pi_{appr}. \tag{4}$$

---

[*]Corresponding author

Clearly, $\pi_{appr}$ is upper bounded by $\|\pi_{appr}\|_\infty \leqslant 1$. Therefore, when $\alpha \to 0$, $\pi_{appr}\tilde{\mathbf{P}} - \pi_{appr} \to 0$. The proof is concluded. $\quad\square$

**THEOREM 2.** *Based on the definitions in the paper, given an input graph $\mathcal{G}$, when teleport probability $\alpha \to 1$, $\mathbf{\Pi}_{appr} \to \frac{1}{n} \cdot \mathbf{I}^{n \times n}$ and the approximate digraph Laplacian converges as $\mathcal{L}_{appr} \to \mathbf{I} - \frac{1}{2}(\tilde{\mathbf{P}} + \tilde{\mathbf{P}}^T)$, which is a trivial-symmetric Laplacian matrix. Specially, if $\mathcal{G}$ is undirected, then the approximate digraph Laplacian converges as $\mathcal{L}_{appr} \to \mathbf{I} - \mathbf{D}^{-1}\mathbf{A}$, which is a random-walk normalized Laplacian.*

*Proof.* From the Equation 2 above, we obtain

$$\alpha \pi_{appr}\mathbf{1}^{n \times 1} = \pi_\xi, \tag{5}$$

and $\pi_{ppr} = (\pi_{appr}, \pi_\xi)$ is the stationary distribution of $\mathbf{P}_{ppr}$, thus, $\pi_{appr}\mathbf{1}^{n \times 1} = 1 - \pi_\xi$ and

$$\pi_\xi = \frac{\alpha}{1 + \alpha}. \tag{6}$$

Then, we have

$$(1 - \alpha)\pi_{appr}\tilde{\mathbf{P}} + \frac{1}{n}\frac{\alpha}{1+\alpha}\mathbf{1}^{1 \times n} = \pi_{appr}. \tag{7}$$

Therefore,

$$
\begin{aligned}
\frac{\pi_{appr}}{\|\pi_{appr}\|_1} &= \frac{(1 - \alpha)\pi_{appr}\tilde{\mathbf{P}} + \frac{1}{n}\frac{\alpha}{1+\alpha}\mathbf{1}^{1 \times n}}{1 - \pi_\xi} \\
&= \frac{(1 - \alpha)\pi_{appr}\tilde{\mathbf{P}} + \frac{1}{n}\frac{\alpha}{1+\alpha}\mathbf{1}^{1 \times n}}{\frac{1}{1+\alpha}} \\
&= (1 - \alpha)(1 + \alpha)\pi_{appr}\tilde{\mathbf{P}} + \frac{1}{n}\alpha\mathbf{1}^{1 \times n}
\end{aligned}
\tag{8}
$$

Since $\pi_{appr}$ is stationary distribution and $\tilde{\mathbf{P}}$ is transition matrix, $\|\pi_{appr}\tilde{\mathbf{P}}\|_\infty \leqslant \|\pi_{appr}\|_\infty\|\tilde{\mathbf{P}}\|_\infty \leqslant 1$. It is easy to show when $\alpha \to 1$, $\frac{\pi_{appr}}{\|\pi_{appr}\|_1} \to \frac{1}{n}\mathbf{1}^{1 \times n}$ and $\mathbf{\Pi}_{appr} = \frac{1}{\|\pi_{appr}\|_1}\mathrm{Diag}(\pi_{appr}) \to \frac{1}{n} \cdot \mathbf{I}^{n \times n}$. Besides, for $\mathcal{L}_{appr}$ as follows

$$\mathcal{L}_{appr} \approx \mathbf{I} - \frac{1}{2}\left(\mathbf{\Pi}_{appr}^{\frac{1}{2}}\tilde{\mathbf{P}}\mathbf{\Pi}_{appr}^{-\frac{1}{2}} + \mathbf{\Pi}_{appr}^{-\frac{1}{2}}\tilde{\mathbf{P}}^T\mathbf{\Pi}_{appr}^{\frac{1}{2}}\right), \tag{9}$$

when $\alpha \to 1$,

$$
\begin{aligned}
\mathcal{L}_{appr} &\to \mathbf{I} - \frac{1}{2}\left(\frac{1}{\sqrt{n}}\tilde{\mathbf{P}}\sqrt{n} + \frac{1}{\sqrt{n}}\tilde{\mathbf{P}}^T\sqrt{n}\right) \\
&\to \mathbf{I} - \frac{1}{2}\left(\tilde{\mathbf{P}} + \tilde{\mathbf{P}}^T\right)
\end{aligned}
\tag{10}
$$

Meanwhile, in the case that $\mathcal{G}$ is undirected, $\tilde{\mathbf{P}}$ is symmetric and $\tilde{\mathbf{P}} = \tilde{\mathbf{P}}^T = \tilde{\mathbf{D}}^{-1}\tilde{\mathbf{A}}$, $\mathcal{L}_{appr}$ coverages to random walk form of $\tilde{\mathbf{A}}$:

$$\mathcal{L}_{appr} \to \mathbf{I} - \tilde{\mathbf{D}}^{-1}\tilde{\mathbf{A}}. \tag{11}$$

We find that $\mathcal{L}_{appr}$ converges to the random walk form $\mathbf{I} - \tilde{\mathbf{D}}^{-1}\tilde{\mathbf{A}}$ of graph $\mathcal{G}$. The proof is concluded. $\quad\square$

# 3 Reproducibility Details

To support the reproducibility of the results in this paper, we detail datasets, the baseline settings pseudocode and model implementation in experiments. We implement the DiGCN and all the baseline models using the python library of PyTorch [1], Pytorch-Geometric [2] and DGL 0.3 [2]. All the experiments are conducted on a server with one GPU (NVIDIA RTX-2080Ti), two CPUs (Intel Xeon E5 * 2) and Ubuntu 16.04 System.

## 3.1 Datasets Details

We use four open access datasets to test our method. Label rate is the fraction of nodes in the training set per class. We use 20 labeled nodes per class to calculate the label rate.

Table 1: Datasets Details

| *Datasets* | Nodes | Edges | Classes | Features | Label rate |
|---|---|---|---|---|---|
| CORA-ML | 2995 | 8416 | 7 | 2879 | 4.67% |
| CITESEER | 3312 | 4715 | 6 | 3703 | 3.62% |
| AM-PHOTO | 7650 | 143663 | 8 | 745 | 2.10% |
| AM-COMPUTER | 13752 | 287209 | 10 | 767 | 1.45% |

## 3.2 Baselines Details and Settings

The baseline methods are given below:

• **ChebNet** [1]: It redefines graph convolution using Chebyshev polynomials to remove the time-consuming Laplacian eigenvalue decomposition.

• **GCN** [4]: It has multi-layers which stacks first-order Chebyshev polynomials as graph convolution layer and learns graph representations use a nonlinear activation function.

• **SGC** [11]: It removes nonlinear layers by using 2-hop adjacency matrix as replacement and collapse weight matrices to reduce computational consumption.

• **APPNP** [5]: It utilizes a new propagation scheme based on *personalized* PageRank to handle feature oversmoothing problem.

• **InfoMax** [10]: It relies on maximizing local mutual infomation and works in an unsupervised way which can be applied to both transductive and inductive learning setups.

• **GraphSage** [3]: It proposes a general inductive framework that can efficiently generate node embeddings for previously unseen data

• **GAT** [9]: It applies attention mechanism to assign different weights to different neighborhood nodes based on node features.

• **DGCN** [8]: It combines first- & second-order proximity matrices together to learn directed features.

• **SIGN** [6]: It proposes a Inception-like structure which uses SGC as basic block and concatenate these block of different size together into a FC layer.

For all baseline models, we use their model structure in the original papers, including layer number, activation function selection, normalization and regularization selection, etc. It is worth noting that GraphSage has three variants in the original article using different aggregators: **mean**, **meanpool** and **maxpool**. In this paper, we use **mean** as its aggregator since it performs best [7]. Detailed hyper-parameter settings are shown in Table 2.

Table 2: The hyper-parameters of baselines.

| Model | layers | lr | weight-decay | dropout | hidden dimension | Others |
|---|---|---|---|---|---|---|
| ChebNet | 2 | 0.01 | 5e-4 | 0.5 | CORA-ML & CITESEER:16 others:64 | num-hop=2 |
| GCN | 2 | 0.01 | 5e-4 | 0.5 | 64 | - |
| SGC | 1 | 0.1 | 5e-4 | 0.5 | - | power-times=2 |
| APPNP | 2 | 0.01 | 5e-4 | 0.5 | 64 | $\alpha = 0.1$ |
| InfoMax | 1 | 0.001 | 5e-4 | 0 | 2048 | max-LR-iter=150 |
| GraphSage | 2 | 0.005 | 5e-4 | 0.6 | CORA-ML & CITESEER:16 others:64 | mean |
| GAT | 2 | 0.005 | 5e-4 | 0.6 | CORA-ML & CITESEER:8 others:32 | heads=16 |
| DGCN | 2 | 0.01 | 5e-4 | 0.5 | 64 | concatenation |
| SIGN | 2 | 0.1 | 5e-4 | 0.5 | 64 | $k = 2$ |

### 3.3 Pseudocode

---

**Algorithm 1:** Digraph convolution procedure

---

**Input:** Adjacency matrix: $\mathbf{A}$, features matrix: $\mathbf{X}$, teleport probability $\alpha$, learnable weights: $\boldsymbol{\Theta}$
**Output:** Convolution result $\mathbf{Z}$

1   **Initialize $\boldsymbol{\Theta}$**;
2   $\tilde{\mathbf{A}} \leftarrow \mathbf{A} + \mathbf{I}^{n \times n}$ ;
3   $\tilde{\mathbf{P}} \leftarrow \tilde{\mathbf{D}}^{-1}\tilde{\mathbf{A}}$ ;
4   $\mathbf{P}_{ppr} \leftarrow \text{AddAuxNode}(\tilde{\mathbf{P}}, \alpha)$;
5   $\pi_{ppr} \leftarrow \text{LeftEVD}(\mathbf{P}_{ppr})$;
6   $\pi_{appr} \leftarrow \pi_{ppr}(1:n)$;
7   $\boldsymbol{\Pi}_{appr} \leftarrow \frac{1}{\|\pi_{appr}\|_1}\text{Diag}(\pi_{appr})$;
8   $\mathbf{Z} \leftarrow \frac{1}{2}\left(\boldsymbol{\Pi}_{appr}^{\frac{1}{2}}\tilde{\mathbf{P}}\boldsymbol{\Pi}_{appr}^{-\frac{1}{2}} + \boldsymbol{\Pi}_{appr}^{-\frac{1}{2}}\tilde{\mathbf{P}}^T\boldsymbol{\Pi}_{appr}^{\frac{1}{2}}\right)\mathbf{X}\boldsymbol{\Theta}$;
9   **return $\mathbf{Z}$**

---

---

**Algorithm 2:** DiGCN procedure

---

**Input:** Adjacency matrix: $\mathbf{A}$, features matrix: $\mathbf{X}$, width of Inception block $k$, fusion function $\Gamma$;
activatation function: $\sigma$, teleport probability $\alpha$, learnable weights: $\{\boldsymbol{\Theta}^0, \boldsymbol{\Theta}^1, ..., \boldsymbol{\Theta}^k\}$
**Output:** Convolution result $\mathbf{Z}_{\mathcal{I}}$

1   **Initialize $\{\boldsymbol{\Theta}^0, \boldsymbol{\Theta}^1, ..., \boldsymbol{\Theta}^k\}$**;
2   **for** $i \leftarrow 0$ to $k$ **do**
3     **if** $i = 0$ **then**
4       $\mathbf{P}^{(i)} \leftarrow \mathbf{I}^{n \times n}$;
5     **end**
6     **else if** $i = 1$ **then**
7       $\tilde{\mathbf{A}} \leftarrow \mathbf{A} + \mathbf{I}^{n \times n}, \tilde{\mathbf{D}} \leftarrow \text{RowSum}(\tilde{\mathbf{A}}), \mathbf{P}^{(i)} \leftarrow \tilde{\mathbf{D}}^{-1}\tilde{\mathbf{A}}$;
8     **end**
9     **else**
10       $\mathbf{P}^{(i)} \leftarrow \text{Intersect}((\mathbf{P}^{(1)})^{k-1}(\mathbf{P}^{(1)^T})^{k-1}, (\mathbf{P}^{(1)^T})^{k-1}(\mathbf{P}^{(1)})^{k-1})/2$;
11       $\mathbf{W}^{(i)} \leftarrow \text{RowSum}(\mathbf{P}^{(i)})$;
12     **end**
13   **end**
14   **for** $j \leftarrow 0$ to $k$ **do**
15     **if** $j = 0$ **then**
16       $\mathbf{Z}^{(0)} \leftarrow \mathbf{X}\boldsymbol{\Theta}^{(0)}$;
17     **end**
18     **else if** $j = 1$ **then**
19       $\mathbf{Z}^{(1)} \leftarrow \text{DigraphConv}(\mathbf{P}^{(1)}, \alpha)$;
20     **end**
21     **else**
22       $\mathbf{Z}^{(j)} \leftarrow \mathbf{W}^{(j)^{-\frac{1}{2}}}\mathbf{P}^{(j)}\mathbf{W}^{(j)^{-\frac{1}{2}}}\mathbf{X}\boldsymbol{\Theta}^{(j)}$;
23     **end**
24   **end**
25   $\mathbf{Z}_{\mathcal{I}} \leftarrow \sigma(\Gamma(\mathbf{Z}^{(0)}, \mathbf{Z}^{(1)}, ..., \mathbf{Z}^{(k)}))$;
26   **return $\mathbf{Z}_{\mathcal{I}}$**

---

## 3.4   Implementation for Digraph Semi-supervised Node Classification

In this section, we implement our model to solve digraph semi-supervised node classification task. More specifically, how to mine the similarity between node class using adjacency matrix $\mathbf{A}$ when there is no graph structure information in node feature matrix $\mathbf{X}$. We are ready to define our task.

**DEFINITION 2.** *Digraph Semi-Supervised Node Classification. Given a digraph $\mathcal{G} = (\mathcal{V}, \mathcal{E})$ with adjacency matrix $\mathbf{A}$, and node feature matrix $\mathbf{X} \in \mathbb{R}^{n \times c}$, where $n = |\mathcal{V}|$ is the number of nodes and $c$ is the feature dimension. Given a subset of nodes $\mathcal{V}_l \subset \mathcal{V}$, where nodes in $\mathcal{V}_l$ have observed labels and generally $|\mathcal{V}_l| << |\mathcal{V}|$. The task is using the labeled subset $\mathcal{V}_l$, node feature matrix $\mathbf{X}$ and adjacency matrix $\mathbf{A}$ predict the unknown label in $\mathcal{V}_{ul} = \mathcal{V} \smallsetminus \mathcal{V}_l$.*

For this task, we first build a two layer network model on digraphs which only use digraph convolution. We schematically show the model in Figure 1 and use DiGCL to represent digraph convolution layer. Our model can be written in the following form of forward propagation:

$$
\begin{aligned}
\hat{\mathbf{A}} &= \frac{1}{2}\left( \mathbf{\Pi}_{appr}^{\frac{1}{2}} \tilde{\mathbf{P}} \mathbf{\Pi}_{appr}^{-\frac{1}{2}} + \mathbf{\Pi}_{appr}^{-\frac{1}{2}} \tilde{\mathbf{P}}^T \mathbf{\Pi}_{appr}^{\frac{1}{2}} \right) \\
\mathbf{Y} &= \text{softmax}\left( \hat{\mathbf{A}}(\text{ReLU}(\hat{\mathbf{A}}\mathbf{X}\mathbf{\Theta}^{(0)})\mathbf{\Theta}^{(1)} \right)
\end{aligned}
\qquad . \qquad (12)
$$

Figure 1: The schematic depiction of model using only digraph convolution. Model inputs are the adjacent matrix $\mathbf{A}$ and features matrix $\mathbf{X}$, while outputs are labels of predict nodes $\mathbf{Y}$.

Moreover, we build DiGCN model using $k^{th}$-order proximity as Inception block, which can be written in the following form of forward propagation:

$$
\begin{aligned}
\mathbf{Z}_{\mathcal{I}}^{(l)} &= \sigma(\Gamma(\mathbf{Z}^{(0)}, \mathbf{Z}^{(1)}, ..., \mathbf{Z}^{(k)})^{(l-1)}) \\
\mathbf{Y}_{\mathcal{I}} &= \text{softmax}\left( \mathbf{Z}_{\mathcal{I}}^{(l)} \right)
\end{aligned}
\qquad , \qquad (13)
$$

where $l \in \mathbb{Z}^+$ is the number of layers and $(\cdot)^{(l)}$ represents the weights of $l^{th}$ layer. The activation function $\sigma$ and the fusion function $\Gamma$ are chose differently according to the experiments in the paper. We show the model in Figure 2.

Figure 2: The schematic depiction of DiGCN for semi-supervised learning. Model inputs are an adjacent matrix $\mathbf{A}$ and a features matrix $\mathbf{X}$, while outputs are labels of predict nodes $\mathbf{Y}_{\mathcal{I}}$.

We do grid search on the hyperparameters: lr in range [0.001, 0.1], weight decay in range [1e-5, 1e-3] and dropout rate in range [0.3,0.8] and use all labeled examples to evaluate the cross-entropy error for semi-supervised node classification task. The val accuracy on CORA-ML and AM-PHOTO with the number of layers and hidden dimension are shown in the Figure 3(a,b) respectively. Detailed hyper-parameter settings of out models are shown in Table 3.

|          (a) Cora-ML          |          (b) Am-Photo          |

Figure 3: Val accuracy on CORA-ML and AM-PHOTO

Table 3: The hyper-parameters of our models. "w/ pr" means digraph convolution using $\mathcal{L}_{pr}$; "w/ appr" means digraph convolution using $\mathcal{L}_{appr}$; "w/o IB" means using $1^{st}$-order proximity digraph convolution only; "w/ IB" means using Inception block.

| Our models     | layers | lr   | weight-decay | dropout | hidden dimension | Others             |
|----------------|--------|------|--------------|---------|------------------|--------------------|
| w/ pr    w/o IB | 2      | 0.05 | 1e-4         | 0.5     | 64               | $\alpha = 0.1$     |
| w/ appr w/o IB | 2      | 0.05 | 1e-4         | 0.5     | 64               | $\alpha = 0.1$     |
| w/ appr w/  IB | 3      | 0.01 | 5e-4         | 0.6     | 32               | $\alpha = 0.1, k = 2$ |

## 3.5   Implementation for Link Prediction Task in Digraphs

We implement the link prediction task using the Pytorch-Geometric [1]. The ratio of positive validation edges is 0.05 and the ratio of positive test edges is 0.1. We do not use early stop and obtain the mean and std that are calculated for 20 random dataset splits and a maximum number of epochs of 500.

Table 4: The hyper-parameters of link prediction models.

| Models | layers | lr   | hidden dimension | output dimension | activation function | Others         |
|--------|--------|------|------------------|------------------|---------------------|----------------|
| GCN    | 2      | 0.01 | 128              | 64               | ReLU                | -              |
| DiGCN  | 2      | 0.01 | 128              | 64               | ReLU                | $\alpha = 0.1$ |

## Footnotes

[1]https://pytorch.org

[2]https://www.dgl.ai

[1] https://github.com/rusty1s/pytorch_geometric

[8] Z. Tong, Y. Liang, C. Sun, D. S. Rosenblum, and A. Lim, "Directed graph convolutional network," *arXiv preprint arXiv:2004.13970*, 2020.

[9] P. Veličković, G. Cucurull, A. Casanova, A. Romero, P. Lio, and Y. Bengio, "Graph attention networks," *arXiv preprint arXiv:1710.10903*, 2017.

[10] P. Veličković, W. Fedus, W. L. Hamilton, P. Liò, Y. Bengio, and R. D. Hjelm, "Deep graph infomax," *arXiv preprint arXiv:1809.10341*, 2018.

[11] F. Wu, T. Zhang, A. H. d. Souza Jr, C. Fifty, T. Yu, and K. Q. Weinberger, "Simplifying graph convolutional networks," *arXiv preprint arXiv:1902.07153*, 2019.