[Reviews · NeurIPS 2020]

Review 1

Summary and Contributions: This paper makes two valuable contributions. First, to enbale spectral GCN to handle directed graphs, it starts from Eq. (1) raised by [8] and raises its weakness in high computation complexity, then it proposes personalized PageRank by introducing an auxiliary node to keep aperiodic and irreducible properties while retaining the sparsity of the adjacency. The authors proved two theorems: by controlling the value of pernality \alpha, the proposed form will degenerate to trivial-symmetric Laplacian and random-walk normalized one. Nice story and rigorous analysis. Second, to enable larger receptive field, the authors further extend the digraph convolution to multiple scales and develop an inception module for the implementation.

Strengths: 1. Well wirtten paper. I enjoy reading it. 2. Performing digraph convolution by Eq. (3) is valuable and novel. The conclusions by Theorem 1 and 2 are simple but meaningful. 3. Instead of applying the k-order adjacency matrix directly, this paper defines the k-order proximity matrix that better suits the structure of directed connections. This consideration is reasonable. 4. Experiments are good and sufficient.

Weaknesses: 1. One potential weakness is the insufficient introduction of [8]: why Eq. (1) works better than simple symmetric form in Line 78? Can Eq. (1) exhibit more desired properties? Perhaps more specifications are needed. 2. Why does intersect in Line 168 mean? does it reflect that both meeting and diffusion paths meet? If so, make it clear. The definitions of the k-order proximity matrix are quite different for k=1 and k>1 in Table 1, why?

Correctness: This paper is well claimed and well supported by its empirical results.

Clarity: Well written paper. Some minor specifications are needed (see above).

Relation to Prior Work: The differences from previous papers are generally well justified. Yet, I have found two papers that also raise the ideas of high order GCN [A] and inception module [B], which are suggested to be cited and discussed. [A] MixHop: Higher-Order Graph Convolutional Architectures via Sparsified Neighborhood Mixing, ICML 2019. [B] DropEdge: Towards Deep Graph Convolutional Networks on Node Classification, ICLR 2020.

Reproducibility: Yes

Additional Feedback: 1. Figure 1 b is not easy to be understood. More clarifications are suggested. 2. Line 199-200, why? 3. Line 141: removing the comma? 4. It is better to include the results of the model without IB and with simple symmetric adjacency matrix. #### post rebuttal ### The authors' responses confirm my justification. Accept.


Review 2

Summary and Contributions: The paper proposes a convolutional neural network for directed graphs. The network architecture considers node neighbours at different distance, inspired by the inception network.

Strengths: -The problem faced by the paper is interesting and timely. -The proposed approach seems reasonable.

Weaknesses: -No details about model selection are provided

Correctness: The experimental part is weak since authors did not provide details about model selection and the paper discussion suggests that the selection of the model's hyperparameters was performed on the test sets (i.e. the results with maximum average performances over the 20 splits are reported)

Clarity: yes

Relation to Prior Work: yes

Reproducibility: Yes

Additional Feedback: -No details about model selection are provided, i.e. how the network architecture and its hyperparameters were set. -The results provided for the study of the effect of teleport probability and table 4 suggest that in table 3 authors reported the best results on the test set, i.e. they performed model selection in the test set. This would lead to a biased comparison. Authors should clarify this point. -Time and Space Complexity:in the space complexity calculation, you use a practical consideration to derive the big-O space complexity notation. By definition, that notation should consider the worst case complexity. Minor remarks: l61: the consideration about the receptive field is true for all convolutions that consider k-hops neighbours at distance k greater than one. l108 matrix that -> matrix l123 coverage -> converge ---- I update my score after the rebuttal provided by authors, that adequately addressed my main concerns


Review 3

Summary and Contributions: Propose a graph neural network model for directed graphs.

Strengths: The proposed model is novel. The problem is important. Experiments on node classification are conducted.

Weaknesses: The experiments should be improved. More work could be compared.

Correctness: The method design is reasonable.

Clarity: Well written paper.

Relation to Prior Work: More work could be compared.

Reproducibility: No

Additional Feedback: Detailed review. -- Summary The manuscript proposes DiGCN, a graph neural network for directed graphs. The model extends GCN to directed graph using Digraph Laplacian based on PageRank. The Digraph Inception Convolutional Networks is further presented. Experiments on several datasets demonstrate that the proposed model outperforms some baseline methods for node classification. Pros 1 The problem is important. 2 The proposed model is novel. The overall quality of this work is good. 3 Experiments on node classification are conducted. Ablation study and some analytical experiments are also provided. Cons/Questions 1 Experiments could be improved. Only node classification task is conducted. It is better to perform more downstream tasks such as link prediction. In addition, it is better to compare and discuss some recent graph neural network models such as: GMNN: Graph Markov Neural Networks, ICML 2019 Deep Graph Infomax, ICLR 2019 How powerful are graph neural networks, ICLR 2019 2 I also concern about the space complexity of proposed model. The datasets used in this work are relatively small with thousands of nodes while the model faces OOM issue. It is necessary to conduct experiments using larger datasets. To summarize, the novelty and overall quality of this work is good while the experiments could be improved.


Review 4

Summary and Contributions: The authors solve two problems on GCN. The first is that Vanilla GCN can only handle undirected graphs. The authors propose Digraph Laplacian based on PageRank. However, because of the computational overhead, they further reduce the computational time and memory requirement by Approximate Digraph Laplacian based on Personalized PageRank. Second is that the Fixed Receptive Field in Vanilla GCN. The authors propose kth-order Proximity and concatenate different kth-order Proximity into an inception block.

Strengths: 1. This paper provides an approximation Digraph Laplacian with theoretical justification that can save both computational time and memory requirements. 2. The authors claim by combining the above two network structures their work can generalize to State-Of-The-Art models easily.

Weaknesses: 1. The kth-order Proximity is similar to prior work with no significant improvement in this part. Given such, the impact becomes smaller. 2. The performance for its major competitor, DCGN, is significantly lower than what it have reported in the original DCGN paper,. If the original experiment results are used, then the proposed model does not really outperform DCGN. I wouldn't mind upgrading my grade if there is a convincing explanation on this part.

Correctness: Yes, the proofs are theoretically correct and convincing.

Clarity: Yes, the paper is well structure and easy to read. The mathematical notations are clear, and the writing is coherent and smooth.

Relation to Prior Work: Yes, the author made a complete comparison with prior work and listed all the implemental detail between their work and similar previous work.

Reproducibility: Yes

Additional Feedback:

[Author Response · NeurIPS 2020]

We thank all reviewers for the constructive comments and are glad to see overall positive reviews toward this work.

**Response to R1:** Thank you for acknowledging contributions of our work and offering helpful suggestions.

*Weakness (#1):* The symmetric trick is an intuitive way, while our Eq.(1) is more interpretable and can degenerate to the
trick form. Moreover, as mentioned in Line 91-94, our method uses eigenvector matrix to normalize Laplacian which
considers nodes' in & out-connection respectively, while the symmetric trick sums in & out -degree matrix together
to normalize Laplacian. It ignores the directed structure and may lead to degenerated performance. *(#2):* Exactly,
intersection means having both meeting and diffusion paths. And according to the definition in Line 162, when $k = 1$,
we cannot find zero length paths. Thus, transition matrix is used to find directly connected neighbors.

*Suggestion*: Please refer to *Space Complexity* under **R3**. Sorry about merging answers due to page limit. Meanwhile,
we will update the paper to discuss high-order GCN ideas, revise the typos and make figures explanation clearly. We
will also provide baseline results using simple symmetric adjacency matrix in the Supp. Thank you.

**Response to R3:** Thank you for invaluable feedback. *Model Selection:* We apologize that we did not describe all
details of model selection and raised misleading of choosing test/val set due to page limits. For the digraph convolution
models without Inception block, we set the network architectures and hyperparameters same with GCN since they used
similar spectral analysis. For the model with IB, as we mentioned in Line 244-246, we did split the validation set and
carried out the grid search on it. The val accuracy on CORA-ML and AM-COMPUTER with the # of layers and hidden
dimension are shown in the Figure (a,b) respectively. We chose a three layer model and set it hidden dimension to 32.
Our model trended similarly on the val & test sets, causing the misleading that we selected models on the test set. We
also did grid search on the hyperparameters: lr in range [0.001, 0.1], weight decay in range [1e-5, 1e-3] and dropout
rate in range [0.3,0.7]. The network architectures and hyperparameters were presented in the **Supp Section 4** and we
will update the paper with detailed model selection process. *Space Complexity*: Due to the asymmetry of the digraphs
mentioned in Line 151-154, long paths normally exist between a few points and are not bidirectional. Thus, using
$k^{th}$-order proximity will get unbalanced receptive field and introduce computational complexity. Intersection and union
of meeting and diffusion paths both can handle unbalancing problem. We compare the # of edges per Inception block
using intersection and union with different $k$ on two datasets shown in Figure (c) and find that intersection does help
to reduce the memory-consuming. Thus, we report the practical case that space complexity grows linearly with the #
of edges empirically. However, the worst case does exist when the input graph is undirected and strongly connected.
Though it is unsuitable to our model, which mainly treats reducible digraphs, we will modify the space complexity to
$\mathcal{O}(k|\mathcal{V}|^2)$ and add detailed explanations. We will also revise the paper following your remarks. Thank you.

**Response to R5:** We appreciate the detailed and positive comment, which reflects essential contributions of our work.

*Weakness (#1):* We fully agree that additional experiments are needed. We will update the article to add link prediction
task and compare with listed papers. *(#2):* The OOM model is our initial model based on PageRank indeed. We realize
this (see Line 101-104) and further simplify it using *personalized* PageRank. In Table 2, our simplified method not only
solves the OOM problem, but also improves the accuracy. We also plot training time per epoch on random digraphs
with difference size (with $N$ nodes and $2N$ edges) in Figure (d) to show you that our model can handle about 10 million
nodes in one GPU (11GB). Moreover, we will add detailed experiments on large datasets in the Supp. Thank you.

**Response to R6:** We thank you for giving considerate review and the opportunity to explain. *Weakness (#1):* Our
second contribution is proposing Inception module in digraphs to enable a larger receptive field, while $k^{th}$-order
proximity is a part of it. Digraphs have unique features mentioned in Line 151-154 and neither the existing $k$-hop nor
the $k^{th}$-order method defined on undirected graphs can handle them well. Therefore, we redefine $k^{th}$-order proximity
using intersection in digraphs. It can achieve high performance while reduce the computational complexity. *(#2):* We
used the source code from DGCN's authors in our experiments. The differences in the results are due to two main
factors. First, the experimental tasks in the two papers are different. Although DGCN uses digraph datasets, their
inputs are still symmetric adjacency matrices, which is why baselines in DGCN are similar to the original results in
undirected graphs. Our experiments, however, restrict the inputs to asymmetric adjacency matrices in order to measure
their performance in digraphs. Please see Line 227-233 for experimental task and Line 254-257 for analysis. Second,
DGCN concatenates second-order in & out matrix to obtain neighbor features, which could lead to a significant increase
in the # of edges and cause OOM problem. Our approach goes beyond DGCN not only for better performance under
more stringent experimental conditions, but also for better interpretability. DGCN uses simple symmetric matrices for
first-order proximity in its Eq.(8) and does not explain why directed structural features can be obtained. Thank you.

(a)  (b)  (c)  (d)

[Meta-Review · NeurIPS 2020]

The reviewers consider the problem relevant and the authors' approach novel and interesting. Some initial concerns about the authors experiments were mollified in the rebuttal.